# Exploration of Existing Integrated Mental Health and Addictions Care Services for Indigenous Peoples in Canada

**DOI:** 10.3390/ijerph20115946

**Published:** 2023-05-25

**Authors:** Jasmine Wu, Victoria Smye, Bill Hill, Joseph Antone, Arlene MacDougall

**Affiliations:** 1Michael G. DeGroote Medical School, McMaster University, Kitchener, ON N2G 1C5, Canada; 2Arthur Labatt Family School of Nursing, Western University, London, ON N6A 3K7, Canada; 3Biigajiiskaan: Indigenous Pathways to Mental Wellness, Atlohsa Family Healing Services, London, ON N6A 3C2, Canada; 4Parkwood Institute—Mental Health Care Building, St. Joseph’s Health Care London, London, ON N6A 4V2, Canada; 5Department of Psychiatry, Schulich School of Medicine and Dentistry, Western University, London, ON N6A 5C1, Canada; 6Lawson Health Research Institute, London, ON N6A 4V2, Canada

**Keywords:** indigenous health equity, mental health services, cultural safety

## Abstract

Due to the persistent impacts of colonialism, Indigenous peoples of Canada face disproportionate rates of mental health and substance use disorders, which are often insufficiently addressed by Eurocentric ‘mainstream’ mental health and addiction services. The need to better address Indigenous mental health has led to Indigenous mental health integrated care (hereafter integrated care): programs using both Indigenous and Western practices in their care delivery. This research describes the common lessons, disjunctures, and solutions experienced by existing integrated care programs for Indigenous adults across Canada. It reveals the best practices of integrated care for programs, and contributes to the Truth and Reconciliation Commission of Canada’s Calls to Action #20 and #22. This study, co-designed by an Indigenous Knowledge Keeper and Practitioner, explores the programs’ relational processes through interviews with key informants. The data was analyzed in consultation with Indigenous collaborators to highlight Indigenous values and interpretations, and knowledge co-production. In highlighting the complexity of integrated care, study results show the lessons of ‘Real Commitment to Communities and Community Involvement,’ and tensions and disjunctures of ‘Culture as Healing,’ ‘People-focused vs. Practitioner-focused Programs,’ ‘Community-oriented vs. Individual-oriented Programs,’ and ‘Colonial Power Dynamics in Integrated Care.’ The discussion explores why tensions and disjunctures exist, and suggests how to move forward using integrated care’s lessons and the concept of IND-equity. Ultimately, Indigenous-led partnerships are paramount to integrated care because they leverage Indigenous knowledge and approaches to achieve health equity within integrated care.

## 1. Introduction

From settler treaties in the 18th century to the present day, Indigenous peoples of Canada have faced systemic threats to their cultures. Colonial and neocolonial practices in Canada have disenfranchised and marginalized First Nations, Métis, and Inuit peoples through the Indian Act, the Indian Residential School System, the Sixties Scoop in which tens of thousands of Indigenous children were forcibly removed from their families to live with non-Indigenous families, and ongoing racism and land dispossession [1,2]. The resulting cultural discontinuity and oppression in current Canadian society has been associated with higher rates of mental health disorders, such as depression, and substance use disorders, such as alcoholism, among Indigenous peoples compared to non-Indigenous peoples [3,4]. These health burden disparities are exacerbated by the social determinants of health faced by many Indigenous communities, such as unemployment and sub-standard living conditions [5]. Finally, this intersectionality is further manifested in barriers to healthcare, such as accessing treatment for substance use [6].

Another complication is the responsiveness of ‘mainstream’ services to Indigenous peoples’ mental health needs. This problem is demonstrated by the literature outlining the disjuncture between Indigenous epistemologies and the current mental health system. In particular, Indigenous understanding of health as encompassing meaningful physical, emotional, mental, and spiritual experiences is devalued in ‘mainstream’ mental health services, which instead emphasize biomedicine and individualistic treatments [7,8]. Additionally, these services are often grounded in an assimilation ethos—paternalism, authoritarianism, and ethnocentrism—that perpetuates Western superiority and limits Indigenous autonomy [7,9]. Research has also shown the unequal prioritization of mental health needs between the dominant culture and Indigenous peoples, citing the preferential organized response to inquests into non-Indigenous versus Indigenous deaths. Ultimately, ‘mainstream’ mental health and addiction services have biases that impede their ability to meet the needs of Indigenous peoples.

The realization that mental health approaches for Indigenous patients are inadequate has led to a paradigm shift: Indigenous mental health integrated care (hereafter integrated care) services that blend both Indigenous and Western practices in their care delivery [10]. This model stems from Two-Eyed Seeing, which broadly refers to “learning to see from one eye with the strengths of Indigenous knowledges and ways of knowing, and from the other eye with the strengths of Western knowledges and ways of knowing for the benefit of all” [11] (p. 335). Citing “culture as cure,” further research reveals that integrated care, where Western biomedicine is specifically integrated into Indigenous knowledge and approaches through Indigenous-led partnerships, benefits Indigenous peoples by strengthening traditional knowledge and lifestyles which were denied through colonization and assimilation [12]. Furthermore, they show that such Indigenous-led integrated care improves holistic health outcomes, access to care, and care plan adherence among Indigenous peoples.

One explanation for why integrated care improves Indigenous health is cultural safety. Cultural safety is defined as delivering quality care “through changes in thinking about power relationships and patients’ rights” in a way where those who are marginalized decide what feels safe [13] (p. 10). It differs from terms such as cultural competency, which is often referred to as acquiring sufficient knowledge of Indigenous culture, thus suggesting a static level of achievement that focuses on the individual rather than their relationships with others [13]. Likewise, cultural sensitivity is defined as recognizing the differences between cultures, but does not necessarily outline methods to change current structures in order to improve the experiences of those who are marginalized in a way they desire [13].

Thus, cultural safety, when applied to health systems, mandates genuine power-sharing between Indigenous and Western partners, making it the basis of integrated care [14]. Furthermore, as it ensures that the type of care provided is determined by the patient and their community, it prompts health practitioners to be less narrowly focused on only becoming more competent in the patient’s culture [13,15]. Rather, it suggests that when practitioners reflect on how their own cultural biases and privilege create power imbalances between them and the patient, they are better equipped to provide quality care and achieve health equity [13]. Therefore, integrated care may overcome the shortcomings of ‘mainstream’ mental health and addiction services due to its application of cultural safety theory.

### Research Questions

Although there is research surrounding the chasm between Indigenous perspectives and ‘mainstream’ mental health services, the challenges of bringing these two approaches together in integrated care remain to be explored. Thus, we ask the following research question: what are common lessons, disjunctures, and potential solutions when incorporating Indigenous and Western practices in mental health and addiction services for Indigenous peoples across Canada?

Through this research, we hope to provide insight into the process of integration, ultimately facilitating the use of integrated care in future programs and adding to the literature surrounding this type of care. Eventually, our study findings can contribute to the Truth and Reconciliation Commission of Canada’s Call to Action #20 and #22: “[addressing] the distinct health needs of the Métis, Inuit, and off-reserve Aboriginal peoples” and “[effecting] change within the Canadian healthcare system to recognize the value of Aboriginal healing practices and use them in the treatment of Aboriginal patients in collaboration with Aboriginal healers and Elders”, respectively [2] (p. 3).

## 2. Methods

As a preface, I, the first author, must first position myself within my qualitative research to ensure scientific rigour. I recognize that the other stakeholders and I inevitably shape the data and findings with our lenses, thus showing the importance of reflexivity and positionality in this work. To begin, I am a non-Indigenous person of colour and a second-generation immigrant/settler. Moreover, despite being trained in the biomedical sciences through pursuing a Doctor of Medicine degree at McMaster University, my reflexivity has allowed me to engage ethically in this cross-cultural research, not to perpetuate neocolonial harms against Indigenous peoples.

The second author is an associate professor and director of the Arthur Labatt Family School of Nursing at Western University. The goal of her research is to promote health equity, focused on the areas of mental wellness and Indigenous health. Currently, in partnership, she is studying how to promote equity-oriented mental healthcare (cultural safety and trauma- and violence-informed care) within and between mental health institutions and community agencies.

The third author is a Mohawk who has worked in mental health since 1982 in the disciplines of Nursing, Social Work, and Education. Working in partnership with Atlohsa Family Healing Services and St. Joseph’s Health Care, London, he has co-created Indigenous mental health programs, healing spaces, and pathways to mental wellness using the Two-Eyed Seeing approach.

The fourth author, a MSW and RSW, is Haudenosaunee from Oneida Nation of the Thames and the clinical lead for Biigajiiskaan: Indigenous Pathways to Mental Wellness, a program run in partnership between Atlohsa Family Healing Services and Parkwood Institute Mental Health. His mixed ethnicity and lived experience allow him to operate through a “Two-Eyed Seeing” lens to effectively access and leverage both Indigenous and Western Knowledge bases in his work, while centering on concepts of justice, equity, and cultural safety in organizations to improve health outcomes.

The final author is a non-Indigenous psychiatrist, researcher, and ally who has provided a psychiatric service to an Indigenous-led healthcare organization since 2014. She is also part of the founding team of Biigajiiskaan: a mental wellness program and reconciliation initiative that is co-delivered by a faith-based hospital and Indigenous family healing organization that aims to provide accessible, culturally safe, specialized care for Indigenous people with serious mental illness, addictions, and concurrent disorders.

### 2.1. Recruitment and Sample

Programs were recruited by email and/or telephone from programs that were listed on the following pages: ‘Indigenous, First Nations, Inuit, and Métis Resources’ provided by eMentalHealth.ca, ‘Substance use treatment centres for First Nations and Inuit’ provided by the Government of Canada, and ‘Aboriginal Health Access Centres’ provided by Alliance for Healthier Communities [16,17,18]. To be included in the sample (*N* = 35), programs must have mentioned on their websites that they provide mental health and addictions care to Indigenous adults (18+), using both Indigenous and Western practices in their care delivery. If a program did not have its own website, but the program could be found on an online mental health and addictions services database (such as www.directory.drugrehab.ca (accessed on 3 December 2020)), this database must have referred to both Indigenous and Western practices in the program’s care delivery.

While the researchers did not intend to focus on specific Indigenous groups or territories, it occurred that most programs that agreed to participate were located in Western Canada. There was little representation from the Eastern and Northern regions of Canada. This potentially reflects the level of trust programs have based on their geographical location and the westward direction of colonization when engaging in academic research.

### 2.2. Design and Procedure

This qualitative study utilized telephone interviews conducted by the first author that asked key informants (Program Manager, Director of Care, etc.) of recruited programs to share the relational processes of their program’s care delivery via open-ended questions (such as “How are decisions made about the treatment centre/program/etc., model and its evolution?”). One key informant was interviewed per program. The completed interviews were transcribed, coded for common themes, and analyzed inductively.

To ensure that the research aligns with the Indigenous epistemology of relationality (the world being interconnected), the interviews used a narrative inquiry informed by the conversational method [19,20]. For instance, they are open-ended and semi-structured, offering the flexibility to the key informant to share based on their lived and living experiences [19]. This also allowed the interviews to be collaborative and dialogic in nature, as the researcher and key informant contributed to each other’s comments. Furthermore, this informal and conversational atmosphere created a relationship between the researcher, key informant, and research topic [19,20]. Finally, the questions asked and topics discussed during the interviews had a broader purpose of decolonizing the healthcare system, which is a shared goal of the conversational method [19].

Additionally, to leverage Indigenous worldviews within this research, the Indigenous framework of Kaupapa Māori was used to inform us on how to uncover the hidden power relations in society [21]. As this study applied Kaupapa Māori to show how the language used to interact with research participants may inconspicuously perpetuate power imbalances, it also aligned with the reconceptualized critical theory of power proposed by Kincheloe and McLaren in 1994 [22]. Ultimately, care was taken to minimize medicalized language in the interviews for this study. For instance, terms such as “treatments” and “treatment centre” are replaced with “services” and the term used by the program to refer to itself, respectively. Furthermore, by collaborating with an Indigenous Knowledge Keeper/practitioner in the study design, we followed Kaupapa Māori’s need for “communicative and collaborative partnerships” [23] (p. 160).

### 2.3. Analysis

This study used inductive thematic analysis, an iterative process where the data collection and analysis often occur simultaneously and are mutually informing [24]. As such, after conducting an interview, we analyzed it for initial codes that would inform the next interview [25]. This was performed systematically through all interviews and included reflexive writing by the first author as auditable evidence for the study’s trustworthiness [25] (Nowell et al., 2017). Care was taken to derive the themes as closely from the interviewee’s words as possible without imposing any theories or lenses. Once our themes emerged from these initial codes, peer debriefing among the research team allowed us to refine them so that they repeatedly told a cohesive and coherent story [25]. This involvement of other collaborators in the data analysis added yet another level of triangulation to ensure scientific rigour.

The data analysis also involved a consultation group of Indigenous practitioners who occupied the roles of a nurse/social worker and social working at a local integrated care program. After sharing the manuscript and virtually presenting the findings, the researcher facilitated a discussion with these collaborators. Such Indigenous involvement further ensured that our findings produce knowledge that values Indigenous interpretations as equal to Western forms of analysis, an essential component to decolonizing research [26]. Finally, in keeping with the Kaupapa Māori framework’s needs for “recurrent and reciprocal” engagement with communities, the study results were also shared with the participating programs [23] (p. 168). In particular, the manuscript was shared with each key informant with the option of providing feedback; email was the chosen method of communication over a live virtual or in-person presentation due to programs being located across the country in different time zones. Sharing the study findings with the programs and inviting them to voice their opinions contributed to more equitable research practices towards Indigenous communities [26]. Given the lack of financial incentives for key informants and consultants, those who were involved with this project volunteered their time, creating an organic exchange of knowledge.

## 3. Results

### 3.1. Qualitative Data

We completed interviews with nine programs across Canada that included representation from urban, rural, and remote programs; programs specifically servicing First Nations and Inuit peoples; and programs from numerous provinces and territories within Canada. From our inductive thematic analysis, we identified five themes and seven subthemes.

To preface the following findings, our intention was to not pit Indigenous knowledge and practices against Western knowledge and practices; there is a role and space for both. Instead, our goal is to highlight the nuances and power dynamics within integrated care and suggest ways for both knowledge systems to work together responsibly. We begin by presenting lessons from the integrated care programs we interviewed (Theme 1), and then we move on to the tensions and disjunctures (Themes 2–5).

Although there was no attempt to get more interviews due to the COVID-19 pandemic’s incessant burden placed on these programs, which already receive limited support, the relationship between the interviewer and participants was maintained after the study’s completion. For instance, knowledge sharing and collaboration among researchers and participants are continuing in follow-up studies regarding funding within these Indigenous mental health services.

### 3.2. Lessons from Existing Integrated Care Programs

Our data analysis of existing Integrated Care programs’ interviews brought forth the following lessons (see Figure 1).

#### Real Commitment to Communities and Community Involvement

“[Our treatment centre] began in 1976—four First Nations people from X and Y communities met to discuss alcohol problems of our people. They felt that the single biggest problem was the lack of alcohol and drug treatment centres. They organized, developed a proposal and lobbied for funding.”(A4).

Self-determination: five out of nine programs mentioned originating from their community’s needs and goals. Often this self-determination required meetings to assess the causes of alcoholism and addictions in the community. Programs also emerged as the communities noted a lack of accessible mental health services in their geographical region.

“I think one thing that a lot of treatment centres work towards is aftercare. […] Before we’d hear [from clients in therapy], “I don’t want to go back home, I’m not comfortable at home. I’m not comfortable to go back into that environment. […] But it would be nice to have more programs that would take clients coming right from treatment. And maybe they’ll see them for two months and maybe, you know, help them get back onto the job, and help them establish healthy [habits] or move back to school.”(A8).

Continuity of Care: From mentioning detoxification services to aftercare, programs emphasized the need to support clients throughout their journey, thus showing their commitment to the continuity of care. Additionally, with many clients “falling through the cracks” (A1) upon leaving, programs strive to ensure that clients can access Indigenous and Western healing services regardless of their home environment.

“We have a Board of Directors, sometimes I would say—and everyone who’s on our board is in sobriety; some of them have been on a long time, a lot of them are what you would consider Elders.”(A5).

“I am the Treatment Director […] I’m a former client of the [counselling centre]. Back in 2009, they were a reconciliation program because of the Indian Day Schools, the Residential Schools. So I went through the program […] I do like using my story because clients will ask me ‘how did you do it?”(A7).

Meaningful Engagement in Programming: Many programs also showed commitment to their community by giving leadership positions to community members with lived and living experience, and former program clients. Personal experiences with mental health and/or addictions services were often shared during the interview, showing an intersection between personal and professional identity. Interviewees also revealed that sharing their own experiences with clients allowed clients to relate to them and build trust.

“We’ve gotten feedback from clients, staff, and referral agents, from court, personal liaisons […] So we’ve really gathered a lot of feedback over the last few years to see if what we’re doing is working and where our needs are. As time has moved on, so have the needs of the individual.”(A1).

Importance of Evaluation: Constantly collecting and implementing feedback from all stakeholders is crucial to these programs for continuous growth. This allowed programs to understand clients’ past and current context and establish meaningful partnerships. Many programs also highlighted their “open-door policy” (A5, A7) to create a safe and non-judgmental space for clients and staff to voice their opinions.

### 3.3. Tensions and Disjunctures within Integrated Care

Figure 2 summarizes the various tensions and disjunctures experienced by Integrated Care Programs while deliv-ering care.

#### 3.3.1. Culture as Healing

“We are a First Nations treatment Centre serving mostly First Nations people and are grounded in the First Nation’s cultural values of respect and honour for all living things.”(A4).

“But we quickly found out that there was something under all of this [alcoholism and addictions], which is from unresolved trauma and colonization, and not feeling good enough and not feeling like they belong because of the disconnection that happened due to the residential schools and scooping our children and taking away our ceremony and banning our languages and keeping us little tiny plots of land, and not letting us live with the land and function in a way that we normally do.”(A5).

“We find that with the [cultural programming], people can adapt to it more—they understand it better. Because they’re able to connect with it, especially if it’s an Elder talking about parenting and stuff like that. They’re going to adapt to it more if an Elder is speaking rather than someone coming and giving a workshop. That kind of thing. [With the Elders], there’s a lot more respect as well.”(A8).

The use of Indigenous knowledge and values while helping clients understand their past showed the importance of cultural context in care delivery. Moreover, many programs also demonstrated the validity of the Elder who were involved on the Board of Directors or worked directly with clients in an authoritative way. Thus, incorporating culture in program development and leadership was integral to the programs.

“Some communities have even banned some of the cultural ceremonies and practices which were traditionally part of their territory. When individuals from those communities that have banned or outlawed these types of ceremony come to the centre, I notice that they’re really hesitant.”(A1).

“So we try to incorporate our language into some of the programs; it’s really hard because some of [the clients] have a ‘mental block’ into trying to speak our language again. I think in some ways, it has to do with colonization and in some ways, some of the people have almost lost our native tongue, and it’s hard to pronounce the sounds properly.”(A6).

However, there is a layer of complexity due to the shame Indigenous peoples have been taught to feel about their cultures. Thus, clients may be hesitant to engage with the cultural components of the program, and do not ascribe to culture as healing. Our interviews reveal the range of responses towards the incorporation of culture, from hesitancy to acceptance and praise.

“One of the programs that we’ve recently introduced in these past few years is the Circle of Security Parenting Program, but we’ve adapted it to [the community’s] values…”(A2).

“And we’d also have cultural aspects of First Nations: sharing circles, healing circles.”(A9).

“So part of the program we have is that we integrate an “on the land” component by arranging for a retreat outside of [the city]. We use an [equestrian retreat] and we integrate equestrian therapy as well as camping, fire, hikes, those sorts of things to try and mimic what our connection to the land or ‘on the land’ learning would look like in an urban context.”(A3).

Along with the diversity in responses towards cultural programming, one must also acknowledge the diversity in cultural programming itself, thus adding more nuance. For example, while some programs adapted Western frameworks and models to fit their community’s needs, other programs directly used their community’s methods. Finally, other programs have had to innovate based on their particular environment while remaining rooted in Indigenous knowledge.

#### 3.3.2. People-Focused vs. Practitioner-Focused Programs

“[Clients] have the choice to choose the change.” And if they don’t want to change, then they’re stuck. If they make that change, they’ll give themselves wings and look at themselves as birds on a wire.”(A7).

“One of the things people always ask me is ‘how successful is your program?’ and I say ‘tell me what you call success? How are you measuring success?’ I think if someone’s life has improved from walking through our doors, whether they stay an hour, a day, a week or complete our program, if something has improved in their life, then I think for them its successful.”(A5).

A recurring theme in programs was to provide a personalized, strength-based approach. As such, clients were able to determine their own goals and heal from within rather than follow the practitioner’s agenda. This aligned with the harm-reduction approach many programs incorporated; instead of defining the standard of success, programs often allowed their clients to determine their measures of success, such as drinking and using substances in a healthier way rather than becoming completely sober. Thus, these programs approach health through a holistic lens that focuses on experience rather than efficiency [27].

#### 3.3.3. Community-Oriented vs. Individual-Oriented Programs

“We developed this whole program based on treating the whole family; it was never just like take the one person that’s drinking, it was like the whole family was looked at in needing support and help.”(A2).

“We also provide a Colleagues Program for individuals who work with First Nations […] Its focus is addressing the effects of intergenerational trauma and the impact of unresolved trauma and shame on our people.”(A4).

Programs often chose to focus on collectivism and community healing rather than the Western notion of individualism [7]. Furthermore, this approach of helping those beyond the person seeking treatment aligned with the Indigenous epistemology of relationality: the interconnectedness of all people, the natural environment, and the spiritual world [19].

#### 3.3.4. Colonial Power Dynamics in Integrated Care

“The person going through treatment maybe got a low educational level, and the language […] It’s like that can be a barrier where… How do you sort of address that if the understanding is not there?”(A9).

“How do you make [topics like neuroscience] understandable and translate it to people who are just making it day-to-day, people who have been using substances… How do you make them really get it?”(A2).

Inaccessible Medicalized Language and Concepts: When developing programs, a challenge in integrated care is incorporating inaccessible and medicalized language and concepts into the care delivery. This emerges most when practitioners communicate with clients and when clients communicate with Elders. Interviewees cited language and cultural barriers, as well as various levels of Western education, as the main reasons for such misunderstandings. 

“A lot of the psychologists and psychiatrists and social workers for that matter don’t understand what happened to us! […] And so then they try to impose their views like there’s something flawed about us.”(A2).

“Right now, something that I’m actively working on is creating a guidance system for our staff, our clients, our partners on Inuit Qaujimajatuqangit (Inuit Knowledge). […] And I think it’s especially important if you’re in an organization where 100% of staff are not Indigenous.”(A3).

“You know what, I’m always teaching. Especially in circles where there’s settlers. The reason I do that is because sometimes it’s just ignorance and they don’t have a clue […] I truly believe that our people know how to heal.”(A5).

Culturally Unsafe Practitioners: In terms of program execution, programs acknowledged their continuous role in educating staff about Indigenous knowledge and Canada’s colonial legacy. Thus, ignorance continues to be pervasive in integrated care and the broader society, perpetuating the power of Western knowledge in care delivery.

“Always working from a budgetary approach—the way the government does—really does limit us. There’s always more things that we want to do moving forward, there are a lot of great ideas that we have as a team, but we are always limited in terms of dollars.” (A1).

“[The healing centre] operated for a few years, and then we lost funding […] the program completely had to shut down, and it was very very difficult for the community to go through that. It was traumatic to both staff that were working at the healing centre and clients.” (A3).

Funding as an Influencer: The colonial government continues to control funding, which strongly impacts the program’s availability and quality. Thus, even though efforts are being made to strive for genuine power-sharing within care delivery, the governmental funding arrangements create a power imbalance inherent in the healthcare system that perpetuates Indigenous health inequity.

## 4. Discussion

### 4.1. Placing Our Study Findings in Theory and History: Why Do These Tensions and Disjunctures Exist?

When a client enters a program, they carry their lived and living experiences and relationships with them; what they choose to accept or reject in treatment depends heavily on their past. As shown by our theme of ‘Culture as Healing’, clients often heal from the cultural programming that has potentially been adapted to, for instance, the program’s physical environment or to fit within an existing Western framework. However, our interviews also showed the role of internalized oppression in relation to integrated care’s cultural programming. This distrust is supported by authors Gonzales, Simard, Baker-Demaray, and Iron Eyes, who note, “in the most treacherous of forms, we believe the systemic oppression and turn our backs on our own Indigenous cultures or its structures that guide our values, ethics, and inherent mechanisms to attach to our cultural ways of Indigenous peoples” [28]. Furthermore, research shows that internalized oppression is transmitted intergenerationally and contributes to the loss of culture and language that many communities face [29,30].

In contrast, our subtheme of ‘Inaccessible Medicalized Language and Concepts’ showed that clients might also have difficulty understanding and relating to Western approaches, contributing to their distrust in engaging with Western treatments [31]. Because such language and concepts are often only understood through formal Western education, this perpetuates an exclusivity that privileges those with such education. Our interviews revealed that these communication barriers and power dynamics occur among practitioner-client-Elder interfaces and may thus limit the cohesion within integrated care. Notwithstanding the effects of internalized oppression, our interviews generally suggest that clients trusted Elders and Indigenous practitioners more. This could be attributed to the distrust clients have when interacting with culturally unsafe practitioners who impose Western practices as better [32]. Thus, the ability of clients to trust their treatment is complex and depends on their experiences of explicit and internalized oppression.

As cultural safety depends largely on practitioner-client interaction, those who continue to perceive Western knowledge as dominant in medicine and society prevent the cultural safety mandate from being met in integrated care [7,33]. Thus, in theory, the mandate of cultural safety has made progress in creating a genuine power-sharing between the two epistemologies; however, in practice, there may still be the perception of Indigenous knowledge as inferior to Western knowledge. Yet, our interviews reveal that programs strive to leverage Indigenous knowledge and values through prioritizing people-focused and community-oriented programs, and thus holistic approaches, relationality, and community healing. As these programs are Indigenous-led partnerships, with Indigenous knowledge as the foundation of these programs and Western knowledge integrated into the care delivery, they resist the persistent power imbalance between Indigenous and Western knowledge and approaches. While more mainstream mental health services are valuing person-focused and community-oriented, the active efforts of integrated care programs to use these frameworks are extra meaningful because of the demographic they service. Choosing to have the program structure rooted in Indigenous helps to dispel the colonial ideas and “mainstream model of health [which are] so pushed and in your face in the helping fields” [7] (p. 12).

However, an added layer of complexity when incorporating Indigenous practices into integrated care is the unique human resource requirements for cultural programming. For example, Indigenous practitioners are underrepresented in healthcare because becoming one requires lifelong learning from ancestors and immersive personal experiences [34]. In addition, learning to share land-based teachings and hold ceremonies takes more time, space, and spiritual involvement than obtaining an academic degree to become a Western practitioner [27]. The different ways of obtaining Indigenous versus Western knowledge and different values placed on credentials may explain the tensions of why Western knowledge is currently perceived as ‘more official’ or superior in healthcare. Ultimately, the shortage of Indigenous practitioners may also explain the ambiguity in Indigenous ‘cultural programming’ such that adapting Western frameworks to Indigenous values requires fewer human resources than holding ceremonies and land-based learning, thus making it easier to implement.

Finally, along with the lack of human resources, there are colonial power dynamics that result in a lack of funding. Our subtheme of ‘Funding as an Influencer’ was prominent in our interviews in that the amount and distribution timeline of government funding heavily shaped programs’ ability to meet community needs. While this issue affects many health and social services, it is particularly detrimental to Indigenous health services, whose clients face more inequities such as racism, discrimination, and collective historical trauma. Currently, in Canada, the most commonly used funding options between the federal government and Indigenous communities are the block funding agreements (BFA) signed for 3–5 years, and the flexible funding agreements (FFA), which require the government’s permission to move funding between budgetary lines [35]. From these forms of classical contracting, defined as having limited flexibility and often lasting a short duration, stems a disconnection within Indigenous health services; the contracts are short-term, but the need for and distribution of funding is ongoing [36]. The resulting uncertainty of funding contributes to service provision problems such as “operating without knowledge of funding allocations, cash flow difficulties, problems in meeting timelines for the spending of funds, and the effects of uncertainty on planning and operational decision-making, on workforce sustainability and on the quality or volume of service provision” [36] (p. 41). All of these problems are not only disruptive to program staff, but also to the program clients, who see the program as a safe space for healing. Being told that their program may not continue because of uncertain funding is traumatizing to participants and breaks the sense of security and trust they have in their program, even if funding does eventually arrive.

### 4.2. What Is Next? Drawing on Integrated Care’s Lessons to Move towards IND-Equity

Our study reveals that an equity-oriented approach is needed when addressing the tensions and disjunctures in integrated care. Thus, we apply the concept of IND-equity in our solutions. Proposed by Downey in 2020, IND-equity is a model of addressing Indigenous health inequities that steps away from Western individualistic notions of fairness and impartiality, and instead is rooted in self-determination and Indigenous knowledge [37]. Our study shows that programs are working towards IND-equity but that more can be done from an Indigenous allyship perspective.

To begin, the theme of ‘Real Commitment to Communities and Community Involvement’ demonstrates the importance of integrated care being Indigenous-led to allow for self-determination, Indigenous knowledge systems, and a rights-based approach—the latter defined as guided by principles of non-discrimination, participation, and inclusion [38]. Through the ‘Self-determination’ subtheme, programs were able to analyze their community’s needs and generate their own solutions contextualized in their community’s knowledge systems and values. Additionally, our ‘Meaningful Engagement’ subtheme showed that programs cycled their former clients and prioritized local community members for leadership; the inclusion of those with lived and living experience in decision-making processes supports a rights-based approach that empowers those accessing the services to participate without judgement. This is also reflected by the ‘Importance of Evaluation’ subtheme, as all program stakeholders are able to voice their concerns in a safe and non-judgemental manner. Furthermore, the collective decision-making process reflects Indigenous values of “consensus and collaboration” (A3). Finally, Indigenous knowledge systems are revealed by Elders and Knowledge Keepers, ensuring program frameworks reflect their community’s needs and the desire for ‘Continuity of Care.’ Even after clients finish their treatment, programs demonstrate a holistic approach to mental health by striving to support clients in other aspects of their lives and recognizing the other factors—such as home environment—that affect mental health.

However, Downey also explicitly mentions the important role of Indigenous allies in advancing IND-equity through personal education and institutional change [37]. Our study reveals that programs are making efforts to engage with allies; however, our tensions and disjunctures reveal that the challenges still need to be overcome. In terms of educating Indigenous allies and even, as Downey acknowledges, some Indigenous peoples in healthcare roles who have internalized colonialism, our results show that programs consistently offer culturally specific training for new staff about Indigenous cultural safety and trauma- and violence-informed care [37]. However, as shown by the ‘Inaccessible Medicalized Language and Concepts’ and ‘Culturally Unsafe Practitioners’ subthemes, racism is still pervasive in integrated care. Thus, a potential recommendation includes not only providing training, which may seem like a formality, but also experiences for new staff to engage with self-reflection, cultural experiences, etc., deeply. For instance, responses mentioned inviting and encouraging new staff to participate in a ceremony. As one interviewee states: “because the only way you’re really going to know is if you participate. […] Just try and educate and inform and include and, you know, we’re walking this path together.” Culturally specific learning could also begin earlier, such as during healthcare practitioners’ academic studies, in order to reduce the abundance of colonial beliefs that need to be unlearned upon practicing in Indigenous health.

In terms of institutional change, along with transforming the educational system to incorporate more Indigenous teachings, Downey emphasizes that “strategic effort also needs to be fiscally supported and engage an appropriate level of human resources to be effective in implementing real and effective change” [37] (p. 107). Currently, our results show that programs consistently partner with non-Indigenous organizations, such as provincial and federal governments, for supplementary funding, training, and care. However, our ‘Funding as an Influencer’ subtheme shows that classical contracting creates reliance and hinders self-determination. Therefore, a recommendation would be to move from classical contracting to relational contracting, which is defined as the expectation of recurring transactions and the program having greater autonomy over the funds. Oftentimes, funders choose classical contracting over relational contracting because it allows the funder to set stricter accountability measures [36]. For instance, if the program receiving funding does not meet the pre-established criteria, then the funder may desire to withhold funding to keep the program accountable. In doing so, the need for written documentation and accreditation is enforced, with Indigenous health programs particularly scrutinized due to a lack of trust towards their credibility [36]. However, relational contracting holds mutual trust between both parties at its core, thus allowing for more self-enforcing mechanisms of fulfilling accountability measures and for stronger relationships [36]. Therefore, relational contracting helps to reduce the traumatizing service provision problems programs face, and helps them move towards community ownership.

Although the interviews provided detailed findings, the researchers acknowledge that the limitations of the pandemic prevented further study in terms of recruiting a larger sample size. It is important to acknowledge that research requires reciprocity, and that despite our research pivoting virtually, programs we desired to recruit may not have had the resources to engage. At the outset, not many programs met the inclusion criteria because integrated care is a relatively new intervention; thus, our recruitment was further impacted when programs had to close due to the pandemic temporarily. Given the inherently highly demanding work championed by programs, along with the increased burdens of lockdown, we are honoured even to have had these programs participate in the study. The pandemic also presented challenges that prevented an even more in-depth investigation of these programs; for instance, the researchers were unable to conduct site visits to each program for an ethnographic approach. Thus, future directions for this study include observation of each site, and doing so would also overcome the limits of relying solely on self-reported verbal data. Other suggestions for future directions include more relationship-building with the programs for stronger partnerships, and promoting different modalities of presenting such research, for example, through oral storytelling and artistic representations. Ultimately, further in-depth research is needed in this area.

## 5. Conclusions

As integrating Indigenous and Western knowledge into mental health and addictions care for Indigenous peoples becomes more common, it is important to acknowledge the lessons, tensions, and disjunctures experienced by existing integrated care programs. Exploring the complexities within this model is not easy; however, doing so informs future integrated care programs, ultimately helping to address the Truth and Reconciliation Commission of Canada’s Calls to Actions. To conclude, integrated care is a constantly changing practice with tensions, disjunctures, and lessons experienced along the way. It is not yet fully understood or supported, giving the academic community a role in advocating for equity-oriented care. The goal is not to remove resources from other health and social services, but rather to distribute them more equitably to attain IND-equity. This research shows the best practices regarding integrated care, fostering the opportunity to create a knowledge hub within regions across Canada about this type of care. Ultimately, it highlights that Indigenous-led partnerships are paramount to leveraging Indigenous knowledge in order to achieve Two-Eyed Seeing, where Indigenous and Western knowledge are both used and valued, truly “for the benefit of all” [11] (p. 335).

## Figures and Tables

**Figure 1 ijerph-20-05946-f001:**
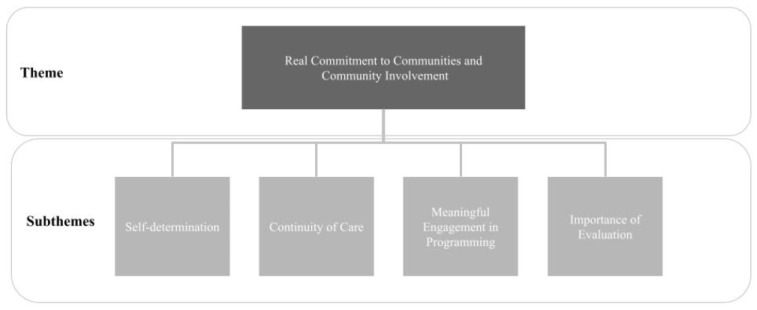
Visual Summary of Lessons from Existing Integrated Care Programs.

**Figure 2 ijerph-20-05946-f002:**
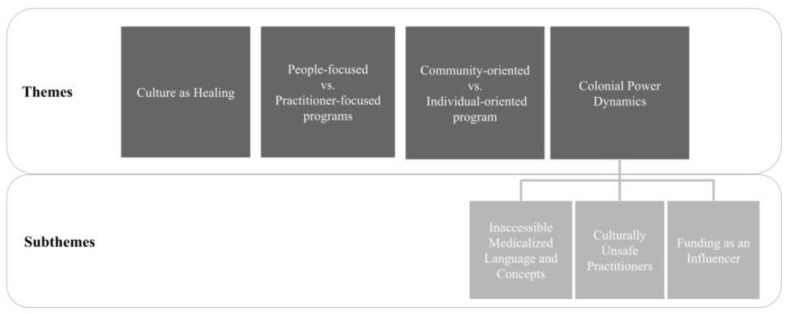
Visual Summary of Tensions and Disjunctures in Integrated Care Programs.

## Data Availability

The data (transcripts) presented in this study are available on request from the corresponding author. The data are not publicly available due to privacy reasons.

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
