# Peer review of "Exploration of Existing Integrated Mental Health and Addictions Care Services for Indigenous Peoples in Canada"

_ijerph, 2023, doi:10.3390/ijerph20115946_

Round 1

Reviewer 1 Report

The manuscript "Exploration of Existing Integrated MH and Addictions Care Services for Indigenous Peoples in Canada" is a mixed-methods study examining integrated care among biomedical and indigenous health and wellness models. The introduction and reflexivity in the methods sections are top notch and could be seen as exemplary ways for other researchers to position themselves and participants within the research. However, the design/procedure and analysis sections need more work and the results section is under-researched. 

Design/Procedure: There is no information on how the survey was created, if it's validated, or information about measures. Also, were there incentives? If not, why? Were the consultants paid? If not, why? Was this approved by an IRB?

Results:

Survey: Only 5 survey's is not nearly enough to show significant power for analysis. Authors state that research is ongoing - consider editing with the other survey's or take out the mixed-methods approach and focus on qualitative only. There are only 3 sentences and a table dedicated to survey results.

Interviews: 9 programs, but how many people? 20 people in 9 programs? 9 people in 9 separate programs? Are the participants patients or managers? I'm assuming managers but I couldn't find that explicitly stated.

Overall: It is mentioned that there was no attempt to get more surveys due to COVID... would incentives had helped? Ethically, how were the programs benefitting from the research? What time frame was the research happening in relation to COVID? This is important since other research projects were able to pivot during COVID (virtual) and able to collect their target numbers. If virtual wasn't possible, tell us why. In the end, I think this is a great paper that either needs to focus on qualitative or go back to the sources and find an avenue to have more quantitative responses. 

Author Response

1) Design/Procedure:

There is no information on how the survey was created, if it's validated, or information about measures. - removed information about the survey based on recommendation.

Also, were there incentives? If not, why? - No incentives were provided based on funding limitations (the study was not funded).

Were the consultants paid? If not, why? - The consultants were not paid based on funding limitations (the study was not funded).

Was this approved by an IRB? - yes, this was already noted in the "Institutional Review Board Statement"

2) Results:

Survey: Only 5 survey's is not nearly enough to show significant power for analysis. Authors state that research is ongoing - consider editing with the other survey's or take out the mixed-methods approach and focus on qualitative only. There are only 3 sentences and a table dedicated to survey results.

- this section has been removed.

Interviews: 9 programs, but how many people? 20 people in 9 programs? 9 people in 9 separate programs? Are the participants patients or managers? I'm assuming managers but I couldn't find that explicitly stated.

- 1 participant from each of the 9 sites. It has been reflected in the methods section that key informants are Program Directors, Managers, etc.

3) Overall: It is mentioned that there was no attempt to get more surveys due to COVID... would incentives had helped? - they could have, but there is value in having the research be a more organic sharing of knowledge through key informants volunteering to participate. Also, this study did not use any funding.

Ethically, how were the programs benefitting from the research? - see 1.1 Research questions. 

What time frame was the research happening in relation to COVID?This is important since other research projects were able to pivot during COVID (virtual) and able to collect their target numbers. If virtual wasn't possible, tell us why. - see lines 609-613.

In the end, I think this is a great paper that either needs to focus on qualitative or go back to the sources and find an avenue to have more quantitative responses. - a qualitative focus was opted for.

Thank you for your time and consideration in reviewing this manuscript.

Reviewer 2 Report

It was a pleasure to review your manuscript. My only comments are:

1. Your manuscript might benefit from a bit stronger emphasis on the limitation imposed by the recruitment process and the small sample size.

2. the next steps might more strongly speak to the need for further in-depth research in this area

Author Response

Hello! Your comments are reflected in the final paragraph before the Conclusions heading. Thank you for your time in reviewing this manuscript.

Reviewer 3 Report

Overall, this is an excellent study using appropriate qualitative methods well. A couple minor points -- maybe Western is the wrong term for conventional medicine for it is done in a similar fashion around the world.  Maybe conventional biomedicine would be better.  I realize it is hard to find the right term.

I would like the authors to speculate about the preponderance of respondents being from Western Canada. I know that there is a program in Thunder Bay, Ontario; Toronto (Anishnabe Health Center), Elsipogtog First Nation in New Brunswick, and Albert Marshall's own reserve in Cape Breton, Nova Scotia.  Are the programs in Eastern Canada afraid to respond or more mistrustful? 

Author Response

1) We have decided to keep "Western biomedicine" to reflect the colonial roots of 'conventional' biomedicine. However, you do bring a really strong point in that this form of medicine is practiced beyond the West.

2) This has been revised on lines 166-167

Round 2

Reviewer 1 Report

Responses are sufficient. No further comments or suggestions.

Author Response

English language and style were revised and spell check updated. The word "cross-sectional" was removed from the manuscript. 
